# PRMT1, a Key Modulator of Unliganded Progesterone Receptor Signaling in Breast Cancer

**DOI:** 10.3390/ijms23179509

**Published:** 2022-08-23

**Authors:** Lucie Malbeteau, Julien Jacquemetton, Cécile Languilaire, Laura Corbo, Muriel Le Romancer, Coralie Poulard

**Affiliations:** 1Université Lyon 1, F-69000, Lyon, France; 2Inserm U1052 CNRS UMR 5286, Cancer Research Center of Lyon, Centre Léon Bérard, F-69008 Lyon, France

**Keywords:** progesterone receptor, PRMT1, transcription, silencing, chromatin

## Abstract

The progesterone receptor (PR) is a key player in major physiological and pathological responses in women, and the signaling pathways triggered following hormone binding have been extensively studied, particularly with respect to breast cancer development and progression. Interestingly, growing evidence suggests a fundamental role for PR on breast cancer cell homeostasis in hormone-depleted conditions, with hormone-free or unliganded PR (uPR) involved in the silencing of relevant genes prior to hormonal stimulation. We herein identify the protein arginine methyltransferase PRMT1 as a novel actor in uPR signaling. In unstimulated T47D breast cancer cells, PRMT1 interacts and functions alongside uPR and its partners to target endogenous progesterone-responsive promoters. PRMT1 helps to finely tune the silencing of responsive genes, likely by promoting a proper BRCA1-mediated degradation and turnover of unliganded PR. As such, PRMT1 emerges as a key transcriptional coregulator of PR for a subset of relevant progestin-dependent genes before hormonal treatment. Since women experience periods of hormonal fluctuation throughout their lifetime, understanding how steroid receptor pathways in breast cancer cells are regulated when hormones decline may help to determine how to override treatment failure to hormonal therapy and improve patient outcome.

## 1. Introduction

Reproductive hormones, mainly produced by the ovaries, fluctuate throughout a woman’s life and are responsible for radical changes at each stage of development [1]. One of the major female reproductive hormones, progesterone, targets the mammary glands via its binding to the progesterone receptor (PR). There, it drives breast physiology from puberty to post-menopause; high levels of progesterone are associated with pregnancy and lactation whereas these decline sharply at menopause [2,3]. Synthetic modulators of PR have been developed to help control different physiological states (like hormonal replacement therapy (HRT) or contraception) and to alleviate medical disorders, from hot flashes to diseases, including endometriosis or cancers [4]. In the latter case, despite a good prognosis of PR-positive breast cancer patients and a better response to endocrine therapy, with a 5-year overall survival superior to 90%, resistance occurs and has driven extensive studies on PR pathway signaling [5]. Indeed, growing data clearly show that initially hormone-responsive ER+/PR+ breast cancers can become hormone independent yet retain functional steroid receptors (SRs) [6]. This transition towards ligand-independent SR activity strengthens the dire need to better understand unliganded SR pathways in hormonotherapy-sensitive tumors but also implies that a drop in circulating hormone levels could be an essential parameter in the emergence of endocrine-resistant tumors in women.

Among the parameters that regulate the activity of steroid receptors, post-translational modifications (PTMs) are crucial. For PR, they have been shown to influence promoter selection, coregulator recruitment, or receptor stability [7], and our team recently unveiled the key role of the protein arginine methyltransferase 1 (PRMT1) in PR transcriptional activity following ligand binding [8]. However, although the regulation of PR pathway after hormonal stimulation has been extensively described, how the receptor is modulated in a ligand-independent context remains largely misunderstood. Scarce studies pointed out that specific gene programs are driven by the hormone-independent form of PR through post-translational modifications such as serine phosphorylation or lysine SUMOylation, both predicted to contribute to breast cancer cell growth and survival [9,10,11]. Furthermore, while most studies described a ligand-dependent shuttling from the cytoplasm to the nucleus to regulate gene transcription, pioneer mechanistic studies also unveiled that, in the absence of ligands, most receptors are constitutively bound to DNA to act as strong repressors of transcription [12]. Vicent and colleagues established a pertinent model in which the unliganded form of PR (uPR) could affect breast cancer progression by regulating gene expression at the chromatin level. They showed that, in the absence of any natural (progesterone) or synthetic (promegestone, R5020) progestins, uPR recruits a repressive multiprotein complex to a subset of responsive promoters to repress basal transcription, maintaining these genes in a silent state prior to hormonal activation [13,14]. 

Here, mostly based on proximity ligation assays (PLAs) and ChIP-qPCRs, we report that PRMT1 interacts within this repressive complex and targets uPR-dependent promoters in hormone-free T47D breast cancer cells. PRMT1 acts as a transcriptional coregulator for a subset of uPR target genes encoding relevant cellular functions, including cell proliferation, and participates in their silencing in the absence of hormonal stimulation. Interestingly, our work herein suggests a gripping link between gene silencing and proteasome degradation of the unliganded receptor prior to hormonal activation. We believe that a better understanding of how PR signaling is controlled when progesterone levels decline is crucial for breast physiology, especially in menopausal women, but will also help to clarify how breast cancer cells respond to modulators used for endocrine therapy.

## 2. Results

### 2.1. PRMT1 Interacts with uPR and Its Partners Prior to Hormonal Treatment

Having recently reported a direct nuclear interaction between PRMT1 and PR prior to hormonal treatment in PR+ T47D breast cancer cells while focusing on ligand-dependent PR signaling [8], we investigated the biological outcome of this association on PR pathway in starved or unstimulated cells (i.e., without any hormonal treatment). In the absence of hormones, the unliganded PR (uPR) binds to genomic sites and targets a multiprotein complex, including the heterochromatin protein 1γ (HP1γ), both histone deacetylases 1 and 2 (HDAC1/2), and the lysine-specific histone H3 demethylase LSD1 [13]. This complex drives the silencing of a subset of endogenous PR-responsive genes involved in key cellular processes (cell proliferation and apoptosis) prior to hormonal stimulation. 

To test whether PR–PRMT1 association is part of this repressive complex, we conducted co-immunoprecipitation (IP) experiments on total proteins extracted from starved T47D cells, cultured in hormone and serum-deprived medium. Using antibodies raised against PR (Figure 1A), PRMT1 (Figure 1B), and HP1γ (Figure 1C), these IP experiments revealed cross interactions between these three proteins. To further validate these results, we conducted in situ proximity ligation assays (PLA) that provided valuable information on the amount and subcellular localization of protein interactions with high sensitivity, in serum-free T47D cells [15,16]. The endogenous interactions between PRMT1, HP1γ, and LSD1 were assessed in pairs and confirmed by the presence of red dots, predominantly in the nuclei (Figure 1D–F). Moreover, a strong decrease in the number of dots was observed when the expression of PRMT1 or HP1γ was knocked down by specific pools of siRNAs (Figure 1G) compared to T47D cells transfected with a control, non-specific siRNA (siNS), validating the specificity of these interactions (Figure 1D–F). Interestingly, PRMT1 also interacted with the H3K9me3 histone in starved cells (Figure 1H), which is a marker of inactive chromatin [17], described to anchor the repressive HP1γ–LSD1 complex in the nucleus of unstimulated T47D cells [13]. The PRMT1/H3K9me3 interaction was nuclear and PRMT1 knockdown led to a significant decrease in the number of dots, indicating a specific PRMT1–H3K9me3 association (Figure 1G). However, since no chromodomain, required for a direct interaction with H3K9me3 [18,19], has so far been identified in PRMT1, we propose that PRMT1 associates within the HP1γ–LSD1–uPR complex with uPR, which then links HP1γ with H3K9me3 and anchors the complex on closed chromatin.

### 2.2. PRMT1 Targets Progesterone-Inducible Promoters in Unstimulated T47D Cells

PRMT1 is known to interact with multiprotein-remodeling complexes on chromatin [20,21] and, in particular, with steroid receptors (SRs) as a potent coregulator [22]. We then investigated whether PRMT1 was recruited to genomic sites regulated by the HP1γ–LSD1–uPR repressive complex. Chromatin immunoprecipitation experiments followed by qPCR analysis (ChIP-qPCR) were conducted in hormone-free T47D cells on three endogenous promoters described to be basally repressed by the HP1γ–LSD1–uPR complex, namely, *STAT5A*, *EGFR,* and *BCL-X* [13]. Firstly, we detected the presence of uPR within these endogenous progesterone-responsive promoters in unstimulated T47D cells (Figure 2A). Secondly, ChIP experiments revealed that PRMT1 targeted the same endogenous progesterone-responsive promoters (Figure 2B). These data support the idea that PRMT1 can be recruited to the uPR complex. 

We then assessed the role of PRMT1 within this repressive complex. No detectable changes in PR recruitment were observed on the progesterone target genes *EGFR* and *STAT5A* upon PRMT1 depletion (Figure 2C), suggesting that PRMT1 does not participate in uPR binding on promoters. However, PRMT1 knockdown differentially influenced the basal expression of these genes; the mRNA level of *STAT5A* increased and those of *EGFR* decreased, while *BCL-X* levels remained unaltered in siPRMT1-transfected T47D cells compared to the control (Figure 2D). As such, PRMT1 targets endogenous progesterone promoters in the absence of hormonal treatment and specifically regulates the expression of these genes, excluding a general effect for PRMT1 on chromatin. 

PRMT1 is a protein arginine methyltransferase catalyzing asymmetric dimethylation on arginine residues on a large variety of substrates, including SRs and PR in particular [23]. We previously showed that PR was not methylated by PRMT1 in T47D cells in the absence of progestin treatment [8], ruling out the influence of a PRMT1-dependent PR methylation on gene expression in starved cells. Here, we confirmed this observation using a selective inhibitor of type I PRMT-dependent methylation, MS 023 [24,25], as the latter had no impact on the basal expression of *EGFR* or *STAT5A* genes (Figure 2E). Hence, PRMT1 is involved in the regulation of the uPR gene expression program, most likely through a mechanism independent of its methyltransferase activity.

### 2.3. PRMT1 Represses the Basal Expression of an R5020-Activated Set of Genes

We previously employed RNA interference and RNA sequencing to identify a subset of progesterone-inducible genes that were regulated by PRMT1 in T47D cells upon hormonal stimulation [8]. T47D cells were depleted of PRMT1 using siRNAs, starved in serum- and hormone-free medium for 72 h, and then stimulated 6 h with R5020 (synthetic progestin) when needed. The efficiency of R5020 stimulation and specificity of PRMT1 targeting were previously demonstrated [8]. Here, we further analyzed these data to assess the relevance and the functional significance of PRMT1 involvement in the regulation of the entire genome without any hormonal stimulation. We used a significant adjusted *p*-value (*p* < 0.05) and a positive fold change without minimal cutoff (Figure 3A,B) to maximize the number of genes included and provide more statistical power for subsequent analyses [26]. Our analysis identified 4081 genes differentially expressed in unstimulated siPRMT1 cells *versus* unstimulated siNS cells (Figure 3A, comparisons 1 and 2; Figure 3B, turquoise circle), highlighting the influence of PRMT1 on gene transcription in T47D cells in the absence of steroid hormones. We concurrently pointed out genes transcriptionally induced after the addition of progesterone and identified 667 genes significantly altered by at least 1 fold after 6 h of the progestin R5020 treatment (Figure 3A, comparison 3; Figure 3B, orange circle). Finally, among them, we identified a list of 156 genes of R5020-induced genes that had a basal expression regulated by PRMT1 (Figure 3B, central red overlap). Surprisingly, while PRMT1 is described as a major SR coactivator after ligand binding, notably by methylating histone 4 on arginine 3 [27,28], most of these genes (60%, 93 genes) were repressed by PRMT1 prior to progestin stimulation (Figure 3C). This result is consistent with our hypothesis that PRMT1 is involved in gene silencing, probably within the uPR silencing complex, prior to any hormonal stimulation and ligand activation of the receptor.

We selected basally regulated genes (*CCND1*, *PDK4*, *GREB1*) encoding critical regulators of breast cancer progression, and performed RT-qPCRs in unstimulated T47D cells. Experiments confirmed that depletion of PRMT1 significantly increased the basal expression of these genes, consistent with whole-genome data (Figure 3D). We strengthened our results with a rescue experiment by expressing a tagged form of rat PRMT1 cDNA (flag-PRMT1 resistant to siRNAs silencing) in siPRMT1-expressing cells. The ectopic expression of PRMT1 rescued the expression of genes affected by PRMT1 depletion in the absence of hormones (Figure 3D), confirming the role of PRMT1 as a gene coregulator. Interestingly, a pathway analysis of the (156) progestin-induced genes basally regulated by PRMT1 revealed an enrichment in genes involved in development and cell growth (Figure 3E). Among them were major regulators of breast cancer cell proliferation, including CCND1, LEF1, KLF9, GREB1, or MAP3K3 [29,30,31,32,33]. In agreement with the pioneer study of Beato’s group [13], these results indicate that, in the absence of hormonal stimulation, PRMT1 takes part in the silencing of a subset of progestin-responsive genes encoding key cellular functions in breast cancer cells.

### 2.4. PRMT1 Is an Interesting Candidate in BRCA1-Dependent uPR Recycling

Having shown that PRMT1 regulates the transcription of PR-dependent genes by facilitating receptor recycling and degradation under hormone stimulating conditions [8], we hypothesized a similar mechanism in the absence of hormone stimulation. Indeed, PRMT1 silencing clearly increased PR protein levels in unstimulated T47D cells (Figure 1H) while regulating a significant number of PR target genes (Figure 3C).

A controlled and rapid SR turnover is a prerequisite for an effective transcriptional activity [34,35,36]. However, studies focusing on PR recycling in the absence of ligand are scarce and to date, only the BRCA1 protein has been described to affect the basal content of uPR [37]. The tumor suppressor BRCA1 (breast cancer and ovarian susceptibility gene 1) gene encodes an E3 ubiquitin ligase that regulates a variety of cellular processes in breast, including DNA repair, transcriptional regulation, or protein ubiquitination, and thus acts as an essential actor in breast cancers [38,39]. Two previous studies used Co-IP to demonstrate that BRCA1 interacts with PR in the absence of progestins in T47D breast cancer cells [37,40]. Using PLA, we showed that PR and BRCA1 mainly interact in the nuclei of starved T47D cells (Figure 4A) and silencing PR or BRCA1 confirmed the specificity of the interaction (Figure 4A,B). Interestingly, knocking down PRMT1 significantly impaired the PR–BRCA1 association (Figure 4A). These results suggest that PRMT1 is important for this interaction and could influence BRCA1-dependent turnover of uPR. Interestingly, in steroid-deprived cells, PRMT1 and BRCA1 knockdown share the same phenotypes, namely, a higher basal level of PR protein in targeted siRNA cells compared to control cells (Figure 4B). As we previously showed that PR is not methylated in the absence of progesterone [8], we proposed that PRMT1 could be a novel actor of PR downregulation prior to hormonal stimulation, acting with BRCA1 in the recycling of unliganded PRs in breast cancer cells.

Our findings raised several hypotheses on the biological significance of this PRMT1–BRCA1 association in starved T47D cells. We speculated that BRCA1 needs to be methylated to better interact with unliganded PR, which is consistent with data showing that PRMT1 methylates BRCA1 independently of hormone stimulation in breast cancer cells, which can affect protein–protein interactions [41] or BRCA1 sub-localization and, therefore, its cellular functions [42]. Additionally, this interaction could be totally independent of arginine methylation, making PRMT1 a scaffold coregulator to recruit BRCA1 independently of its catalytic activity.

## 3. Discussion

Owing to its role in the response of breast cancer patients to endocrine therapy, the progesterone receptor (PR) has been extensively studied in conditions of hormonal stimulation [43]. However, few studies have focused on the unliganded form, either between hormonal stimulations or in hormone-deficient conditions (e.g., post-menopause). A pioneer study unveiled a decisive role for unliganded PR in breast cancer cells by targeting a multiprotein complex to locally close chromatin and regulate basal transcription of PR-responsive genes, probably to avoid any aberrant activation in the absence of ligand [13]. We herein demonstrated that PRMT1 helps in finely tuning the silencing of progesterone-responsive genes prior to hormonal stimulation, acting as a transcriptional coregulator of the unliganded PR form. Mechanistically, we showed that PRMT1 specifically binds to a set of PR target genes encoding key cellular processes (including cell proliferation) in the absence of hormones but also promotes nuclear interactions between uPR and the E3 ubiquitin ligase BRCA1, crucial for the proper and controlled turnover of the receptors. Interestingly, we previously showed that PRMT1 rapidly dissociates from PR upon exposure to its hormonal ligand (15 min) to then be re-associated with liganded PR and methylate it (1 h) when PRMT1 is recruited to promoters, acting as a coactivator of genes encoding cell proliferation and migration [8]. Our observations corroborate previous evidence showing that the repressive HP1γ–LSD1–uPR complex is highly dynamic, being rapidly dissociated and ejected from chromatin only a few minutes after the addition of hormones [13].

Hence, as we observed specific nuclear interactions between PRMT1, uPR, and two major partners of the silencing complex in starved T47D cells (Figure 1), we proposed that PRMT1 associates within the HP1γ–LSD1–uPR and acts as a major player in the silencing of progesterone-responsive genes prior to hormonal stimulation (Figure 4C). Of note, among the 667 progestin-responsive genes we identified by RNA sequencing in T47D cells, a set of 93 genes was repressed by PRMT1 prior to stimulation, representing 14% of the total R5020-activated genes (Figure 3C). This rate is consistent with previous evidence, uncovering an unexpected and significant number of 20% of endogenous PR-responsive promoters repressed in the absence of progestins [13], that controls important cellular processes such as proliferation, development, or cell death. The addition of progestin then leads to the activation of downstream kinase cascades and phosphorylation events, which facilitate complex removal from basally repressed promoters and allow the recruitment of enzymes modifying chromatin [44]. This, in turn, results in local chromatin remodeling and in the formation of the transcription initiation complex on target genes. PRMT1 binds with activated PR, methylating the receptor in order to regulate its stability and its proper transcriptional activation, allowing controlled cell proliferation and migration [8] (Figure 4C).

We herein suggest that PRMT1 inherently links the cyclic turnover of uPR (probably mediated by BRCA1) and the PR-dependent basal gene expression in the absence of hormones. This notion was already investigated for ERs, where the proteasome-dependent degradation of both liganded and unliganded ERα fosters continuous responses to changes in hormone fluctuations in breast cancer cells [45,46]. Moreover, an exciting finding on how unliganded PR can govern *ESR1* expression by regulating DNA methylation was recently published [47]. The binding of uPR to the low-methylated *ESR1* promoter maintains *ESR1* basal expression and thus controls downstream ERα-dependent activity. Conversely, the depletion of PR is associated with increased DNA methylation at the *ESR1* promoter and, therefore, less ERα expression in starved T47D cells. This work suggests that uPR could be determinant for ERα expression and highlights how it can influence the response of cancer cells to estrogen and selective ER modulators (SERM). Indeed, as *PGR* is a target gene of ERα [48,49], endocrine therapies targeting ERα necessarily result in the loss of PR expression. Hence, investigating the role of PR in hormone-free breast cancer cells is turning out to be crucial to understand how cancer cells will respond to treatment. In our work, we investigated the role of PRMT1 in uPR signaling, in a progesterone- and estrogen-deprived context and intentionally omitted the unliganded ERα pathway (to make the analysis of uPR signaling easier to decrypt). However, we are aware that the unliganded PR pathway could also be ER-dependent. ERα is a potent prognostic and predictive marker of endocrine therapy responses for ERα+/PR+ breast tumors [50], and pioneer studies have shown a potent impact of unliganded ERs on breast cancer. Genome-wide ER binding data collected in breast cancer cells growing in hormone-depleted media have underlined specific ligand-independent mechanisms to affect gene transcription, microRNA expression, and downstream proteome [51]. As for PR, ERα binding sites in an unstimulated state represent a non-random set of the cistrome, related to developmental functions [52], strongly supporting a fundamental role for ERα in the homeostasis of breast tumor cells when hormones are ablated physiologically or pharmacologically.

How steroid receptors influence breast cancer initiation and progression when hormones become increasingly rare or receptors become hormone-resistant remains poorly understood. Since circulating levels of progesterone deeply fluctuate throughout life, and even drop following the menopausal transition, we can hypothesize that the silencing of proliferative genes by unliganded PRs and coregulators, including PRMT1, is a crucial process acting as a break to limit the development of breast cancer especially in post-menopausal women. Interestingly, PRMT1 was shown to be dramatically overexpressed in breast cancer, and patients with higher PRMT1 expression display a higher malignancy grade, in part through the activation of proliferation-related genes [53,54]. Our findings raise the idea that a higher expression of PRMT1 in patients, especially at a post-menopausal stage with a hormonal drop, could be associated with PR instability and aberrant expression of proliferative genes, leading to the neutralization of the protective and anti-tumoral effects governed by uPR-dependent gene repression.

However, one main limitation to our further understanding of the complex PR signaling in breast cancer remains the lack of adequate models. In this study, we used the T47D cell line that displays a phenotype particularly useful to dissect the regulatory steps of PR activity. Conflicting and inconsistent reports regarding PR status or progestin response in other luminal A breast cancer cell lines, including MCF-7, make it difficult to work on PR signaling in another established breast cancer cell line [55,56]. This explains why most of the mechanistic studies on this receptor have been exclusively performed on T47D cells, as they express much higher constitutive levels of PR, compared to other breast cancer cells. However, we aim to determine the clinical relevance of our in vitro mechanistic findings in a more physiological preclinical model, notably using mouse and patient-derived xenografts. Further studies are therefore needed to improve our global understanding of the role of unliganded receptors in cancer outcome and to determine how to override treatment failure to hormonal therapy. Our work and that of others raise exciting questions on the biology of unliganded PR in regulating breast tumorigenesis, possibly in connection with ERα signaling, and may help to improve breast cancer patient outcome.

## 4. Materials and Methods

### 4.1. Cell Culture and Treatments

T47D (ATCC) cells were cultured in RPMI-1640 medium, supplemented with 10% fetal bovine serum (FBS), 2% penicillin-streptomycin (Life Technologies, Carlsbad, CA, USA), and insulin (10 µg/mL). They were grown in a humidified atmosphere with 5% CO_2_ at 37 °C, authenticated by Eurofins and tested for Mycoplasma infection by the MycoAlert Mycoplasma Detection Assay (Lonza, Rockland, ME, USA).

Prior to experiments, T47D cells were grown in phenol red-free medium supplemented with 10% charcoal-stripped serum (Biowest, Nuaillé, France). Then, 48 h later, medium was replaced by fresh serum-free medium for 24 h to starve the cells. When indicated, the cells were treated with the MS 023 Type I PRMT inhibitor (Tocris, Bristol, UK) for 48 h at 60 nM (or DMSO vehicle) or with the synthetic progestin R5020 (Perkin Elmer, Waltham, MA, USA) for 6 h.

### 4.2. SiRNA and Plasmid Transfection

The siRNAs’ pool against PR, PRMT1, HP1γ, and BRCA1 and siRNA negative control were purchased from ThermoFisher, Dharmacon, and Eurogentec. SiRNA transfections were performed using Lipofectamine 2000 (ThermoFisher, Waltham, MA, USA) according to the manufacturer’s protocol. After 72 h, the downregulation was analyzed by immunoblot or by RT-qPCR. For rescue experiments, the plasmid-expressing rat PRMT1 [57] was co-transfected with the siRNAs’ pool against PRMT1 using the JetPRIME reagent (Polyplus, Illkirch, France) according to the manufacturer’s protocol.

### 4.3. Immunoprecipitation (IP), Immunoblot (IB), and Antibodies

After starvation, the cells were lysed using RIPA buffer (50 mM Tris HCl pH 8.0, 150 mM NaCl, 1 mM ethylenediamine tetra-acetic acid (EDTA), 1% NP-40, and 0.25% deoxycholate) supplemented with protease inhibitor tablets (Roche Molecular Biochemicals) and phosphatase inhibitors (1 mM sodium fluoride, 1 mM Na_3_VO_4,_ and 1 mM β-glycerophosphate).

A total of 1 mg of protein was incubated with primary antibodies at 4 °C overnight to start the immunoprecipitation (IP). Protein A Agarose beads were added for 2 h at 4 °C on a rotating wheel. Washes were performed with RIPA buffer without inhibitors. Immunoprecipitates and inputs (30 µg) were denatured by boiling in Laemmli sample buffer and analyzed by immunoblot (IB).

The SDS-PAGE gels were electroblotted onto a PVDF membrane and incubated with primary antibodies overnight at 4 °C. Membranes were incubated with horseradish peroxidase (HRP)-conjugated anti-rabbit or anti-mouse immunoglobulins (Jackson Immunoresearch) and proteins were visualized by chemiluminescence (Clarity Western ECL Substrate, BioRad) following the manufacturer’s instructions.

Quantification of the immunoblot band intensity (Figure 4B) was performed with BioRad software. The primary antibodies are listed in Table 1.

### 4.4. Proximity Ligation Assays (PLA), Image Acquisition, and Analysis

PLA assays were performed using the Duolink kit (Sigma-Aldrich, St. Louis, MO, USA) according to the manufacturer’s instructions. T47D cells (3 × 10^5^) were grown on coverslips in medium deprived of steroids for 48 h and then starved in serum-free medium for 24 h before fixation with methanol for 2 min. After saturation in the blocking solution, cells were incubated with primary antibodies at 37 °C for 1 h. The PLA probes consisting of secondary antibodies conjugated to complementary oligonucleotides were then incubated in the same conditions. The step of nucleotides’ ligation (30 min at 37 °C) was followed by the amplification phase, for 100 min at 37 °C in a dark and humidified chamber. Finally, the coverslips were mounted on glass slides in a mounting solution (Dako, Carpinteria, CA, USA) and were analyzed under fluorescence microscopy at 60X magnification (Nikon Eclipse Ni microscope, Amstelveen, The Netherlands).

Image acquisition was performed as described previously [8,58], by imaging DAPI-stained cells at a fixed Z-position with an analysis of Z-sections of ±5 μm at 1 μm intervals. Typically, at least 100 cells were analyzed per condition on 10 random fields of view chosen in an automated manner. The final image was stacked to a single level before further quantification. Analysis and quantification of these samples were performed using the Image J software (Version 1.52, NIH, Bethesda, Rockville, MD, USA).

### 4.5. RNA Extraction and Real-Time qPCR Analysis

RNA extraction was performed on cells grown under starving conditions using TRI-Reagent (Sigma-Aldrich, St. Louis, MO, USA). A total of 1 µg of RNA was reverse transcribed using 100 ng of random primers following the Superscript II (ThermoFisher, Waltham, MA, USA) protocol. Real-time PCR (RT-PCR) was performed with a SYBR Green qPCR master mix (BioRad, Hercules, CA, USA) in a Step One plus real-time PCR detection system (Applied Biosystems, Waltham, MA, USA). Messenger RNA (mRNA) levels were normalized against the level of 28S ribosomal mRNA. Results shown are mean ± SEM for at least three independent experiments. The *p*-value was calculated using a paired *t*-test: * indicates a *p* ≤ 0.05, ** indicates *p* ≤ 0.01, and *** indicates *p* ≤ 0.001. Sequences of the oligonucleotides used in the study are listed in Table 2.

### 4.6. Chromatin Immunoprecipitation (ChIP)

Chromatin was prepared from 5 × 10^6^ starved T47D cells. Cells were crosslinked with 1% formaldehyde (Sigma-Aldrich, St. Louis, MO, USA) for 10 min at room temperature and treated with 0.125 M glycine for 5 min under shaking. Nuclei were lysed in 300 μL of ice-cold RIPA buffer (50 mM Tris HCl pH 8.0, 150 mM NaCl, 1% NP-40, 0.5% NaDoc, and 0.1% SDS) prior to chromatin DNA shearing with a Diogene Bioruptor. The antibody–chromatin complexes were precipitated with salmon sperm DNA/protein A agarose beads for 3 h. Samples were extracted and heated at 65 °C for 5 h to reverse the cross links. After DNA purification, 2 ng of input DNA were used for qPCR analysis. The relative enrichment of a given promoter region obtained with a specific antibody was compared with input DNA, normalized against a reference locus (human chromosome 1 in which no histone modification was reported).

ChIP was performed with the primary antibodies listed in Table 1. Sequences of the primers used to amplify ChIP-enriched DNA are listed in Table 2.

### 4.7. RNA Sequencing and RNA Seq Analysis

The samples and conditions used for the RNA sequencing experiment were described previously [8]. The RNA seq data were submitted to the Gene Expression Omnibus (GEO) and are available using the GSE134194 submission number. Differential gene expression analysis was performed with DEseq2 (Galaxy Version 2.1.8.3) using different thresholds. For siPRMT1-expressing cells in the absence of treatment (eth vehicle), we used: FDR < 0.05, *p*-adjusted value < 0.01, no minimal fold change, and expression > 10 reads per million. For R5020-induced genes in siNS cells, we used: FDR < 0.05, *p*-adjusted value < 0.01, fold change > 1, and expression > 10 reads per million. To obtain the list of the R5020-induced genes with a basal expression regulated by PRMT1, we used: FDR < 0.05, *p*-adjusted value < 0.01, no minimal fold change, and expression > 10 reads per million. To identify the subset of R5020-induced genes repressed by PRMT1 prior to hormonal stimulation, we used: FDR < 0.05, *p*-adjusted value < 0.01, fold change > 0, and expression > 10 reads per million. The experiments were performed three independent times for siRNA transfection and for RNA extraction.

## Figures and Tables

**Figure 1 ijms-23-09509-f001:**
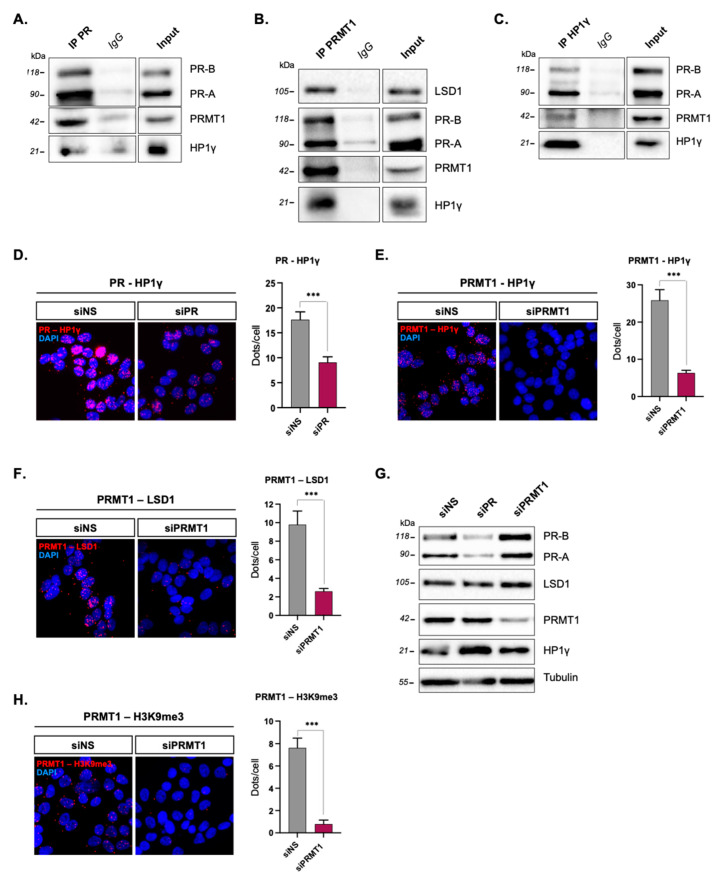
PRMT1 interacts with unliganded PR and partners in nuclei of unstimulated T47D breast cancer cells. T47D cells were cultured in medium deprived of steroids for 48 h and then starved in serum-free medium for 24 h. (**A**–**C**) Whole-cell extracts of T47D were subjected to immunoprecipitation (IP) using (**A**) anti-PR, (**B**) anti-PRMT1, and (**C**) anti-HP1γ antibody or control IgG and immunoblotted (IB) with different antibodies. (**D**–**H**) Proximity ligation assay (PLA) was used to detect the endogenous interactions between (**D**) PR and HP1γ, (**E**) PRMT1 and HP1γ, (**F**) PRMT1 and LSD1, and (**H**) PRMT1 and H3K9me3 in hormone-free T47D cells. T47D cells were transfected with non-silencing control siRNA (siNS) or with anti-PR (siPR) or anti-PRMT1 (siPRMT1) siRNAs; anti-PR, anti-HP1γ, anti-PRMT1, anti-LSD1, and anti-H3K9me3 antibodies were used for PLA assays. The nuclei were counterstained with DAPI (blue) (Obj: X60). The interactions are represented by red dots. The right panels of each figure show the quantification of the number of signals per cell, as described in the Methods section. The mean ± SD of one experiment representative of three experiments is shown. The p-value was determined using the Student’s *t*-test: *** indicates a *p* ≤ 0.001. (**G**) The efficacy of PR and PRMT1 siRNA treatments in unstimulated T47D cells was analyzed by IB. Tubulin was used as a loading control.

**Figure 2 ijms-23-09509-f002:**
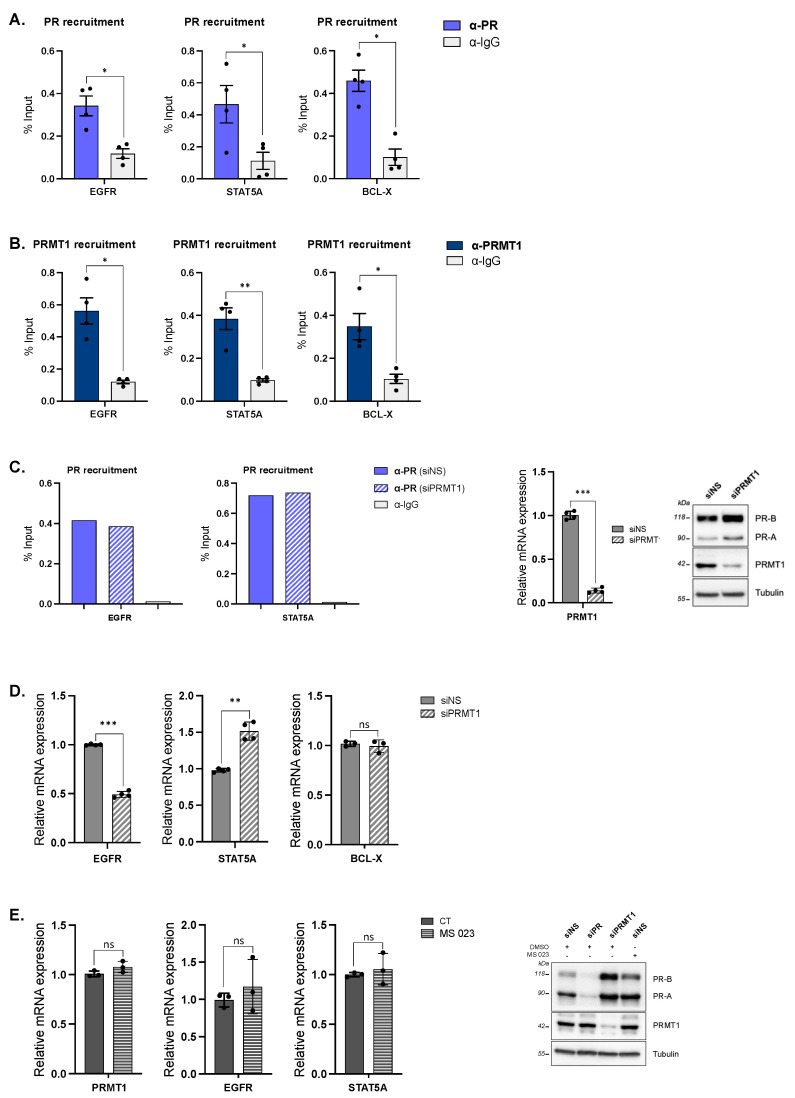
PRMT1 acts as a PR coregulator in the absence of hormones in T47D cells. (**A**–**B**) T47D cells were cultured in medium deprived of steroids for 48 h and then starved in serum-free medium for 24 h. Starved T47D cells were collected and subjected to chromatin immunoprecipitation (ChIP) with (**A**) anti-PR, (**B**) anti-PRMT1 antibodies, or control IgG. The precipitated DNA fragments were used for qPCR analysis using specific primers (*EGFR, STAT5A,* or *BCL-X*) with respect to the input DNA and normalized against a reference locus (human chromosome 1 gene). The p-value was calculated using a paired *t*-test: * indicates *p* ≤ 0.05 and ** *p* ≤ 0.01. The graphs show mean ± SEM of four independent experiments. (**C**) T47D cells, transfected with siNS or siPRMT1, were subjected to ChIP experiment with anti-PR antibody (or control IgG) and the recruitment of PR was analyzed using qPCR on two different target genes (*EGFR*, *STAT5A*), with respect to input DNA and normalized against a reference locus (human chromosome 1 gene). (**Left panel**): The graphs show one experiment representative of two. (**Right panel**): The efficacy of PRMT1 siRNA treatment was analyzed by qPCR (graph) and IB. Tubulin was used as a loading control. (**D**,**E**) Total RNA from T47D cells (**D**) transfected with non-silencing control siRNA (siNS) or with anti-PRMT1 (siPRMT1) siRNAs or (**E**) treated with MS 023 (type I PRMT inhibitor, 60 nM) or vehicle ethanol (CT) was prepared; complementary DNAs (cDNAs) were analyzed by RT-qPCR with specific primers. Mean values were normalized against the expression of 28S ribosomal mRNA as reference. The p-value was calculated using a paired *t*-test: ** indicates *p* ≤ 0.01, *** indicates *p* ≤ 0.001, and ns means “statistically non-significant”. All graphs show mean ± SEM of, at least, three independent experiments. (**Right panel**): IB showing that MS 023 (60 nM) does not affect PRMT1 protein levels. Tubulin was used as a loading control.

**Figure 3 ijms-23-09509-f003:**
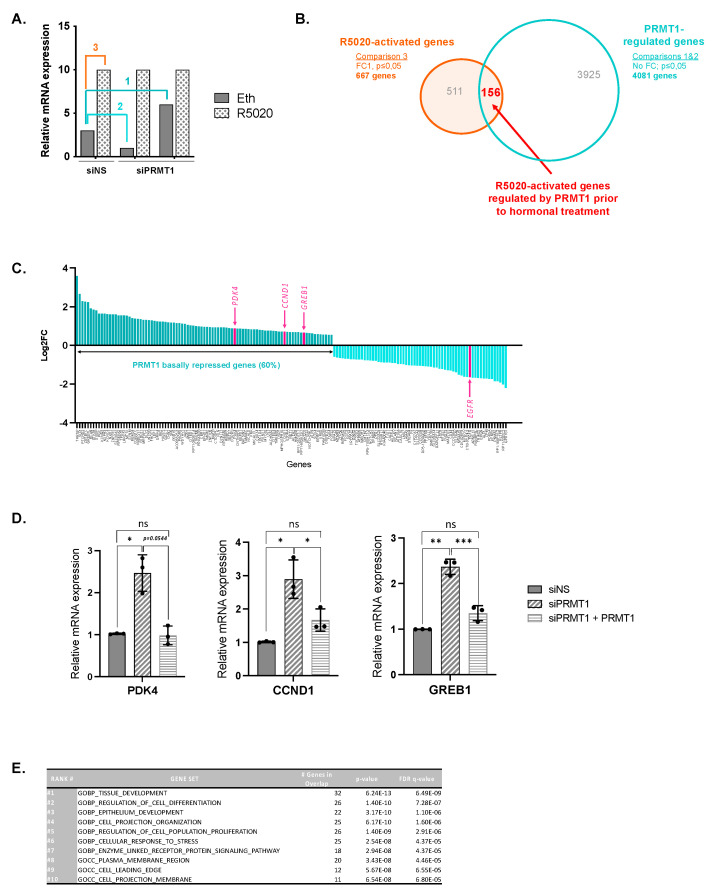
PRMT1 represses the basal expression of an R5020-activated set of genes. (**A**–**C**) Genome-wide RNA sequencing analysis was performed in T47D cells. From previously published data [8], we analyzed the functional impact of PRMT1 on PR target genes prior to any hormonal stimulation. (**A**) Hypothetical graph illustrating how specific pairwise comparisons were performed between datasets for individual samples. (**B**) The blue Venn diagram represents the PRMT1-regulated (overexpressed and downregulated) genes in unstimulated cells (comparisons 1 and 2, respectively); the orange Venn diagram represents the R5020-regulated genes in cells expressing non-specific control siRNA (siNS) (comparison 3). Overlap area (in red) indicates the number of shared genes among sets. Controls for T47D cell treatments can be found here [8]. (**C**) Representation of fold changes (log2FC) of all target gene expression identified by RNA sequencing analysis (156 genes). On the left (dark blue), genes that are downregulated with siPRMT1 (60%) in the absence of hormones. On the right (light blue), genes that are negatively regulated by PRMT1. Several target genes are highlighted (pink) for each group and their expressions were analyzed in (**D**). (**D**) T47D cells were cultured in medium deprived of steroids for 48 h and then starved in serum-free medium for 24 h. Cells were transfected with siPRMT1 (or non-silencing control) and co-transfected with a plasmid expressing a rat PRMT1 to rescue PRMT1 expression. Total RNA was prepared and cDNAs were analyzed by RT-qPCR with the indicated primers. The cDNA mean values were normalized against the expression of 28S ribosomal mRNA as reference. Results shown are mean ± SEM of three experiments. The p-value was calculated using a paired *t*-test: * indicates *p*  ≤  0.05, ** indicates *p*  ≤  0.01, *** indicates *p*  ≤  0.001, and ns means “statistically non-significant”. (**E**) Gene ontology analysis identified PRMT1-regulated gene networks in unstimulated cells. Gene sets are ranked according to their normalized enrichment score (NES). The false discovery rate (FDR) is the estimated probability that a gene set with a given NES represents a false positive.

**Figure 4 ijms-23-09509-f004:**
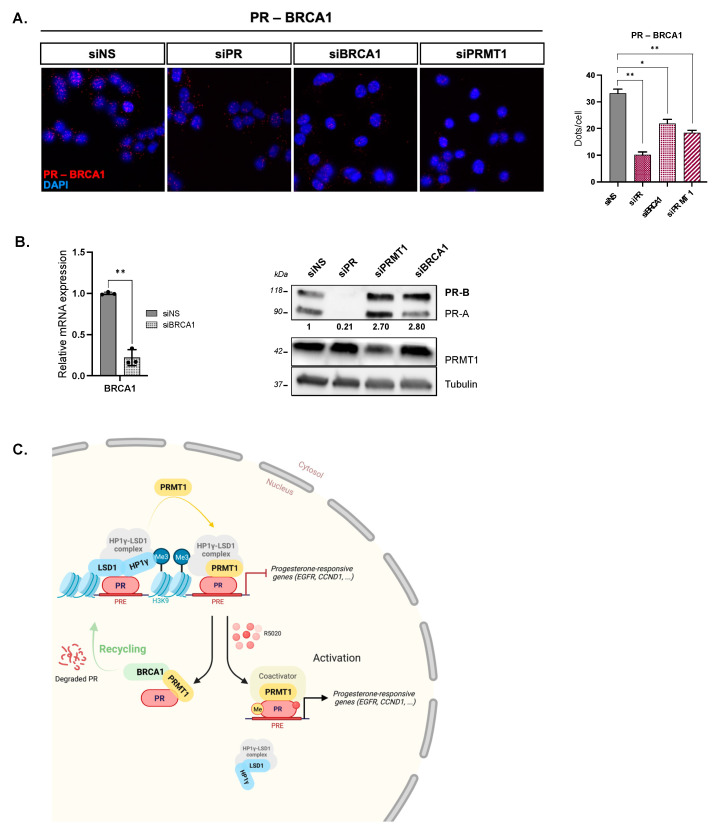
PRMT1 regulates PR–BRCA1 interactions in unstimulated T47D breast cancer cells. T47D cells were cultured in medium deprived of steroids for 48 h and then starved in serum-free medium for 24 h. (**A**) Proximity ligation assay (PLA) was used to detect endogenous interactions between PR and BRCA1 in T47D cells. Cells were transfected with non-silencing control siRNA (siNS) or with anti-PR (siPR), anti-PRMT1 (siPRMT1), or anti-BRCA1 (siBRCA1) siRNAs and anti-PR and anti-BRCA1 antibodies were used. (Left panel): The nuclei were counterstained with DAPI (blue) (Obj: X60). The interactions are represented by red dots. (Right panel): The graph shows the quantification of the number of signals per cell, as described in the Methods section. The mean ± SD of three independent experiments is shown. The p-value was determined using the Student’s *t*-test: * indicates *p*  ≤  0.05 and ** indicates *p*  ≤  0.01. (**B**) The efficacy of the different siRNA treatments in unstimulated T47D cells was analyzed by qPCR (left panel) and IB (right panel). Tubulin was used as a loading control. The ratio (PR-B expression/tubulin expression) was used to quantify PR expression in the different conditions. (**C**) Proposed model: PRMT1 interacts with unliganded PR and some partners as part of a genomic repressive complex regulating the expression of a subset of progestin-responsive genes, including *EGFR*, *STAT5A*, or *CCND1*, in the absence of hormones. In parallel, we also showed that PRMT1 helps unliganded PRs and the E3 ubiquitin ligase BRCA1 to interact. This interaction may be crucial for the proper and controlled turnover of the receptors and, consequently, helps to finely tune the silencing of progesterone-responsive genes prior to hormonal stimulation. The addition of progestin results in removing the repressive complex from basally repressed promoters and allows the recruitment of PR coactivators, including PRMT1, which methylates the receptor to regulate its stability and its proper transcriptional activation. Created with BioRender.com.

**Table 1 ijms-23-09509-t001:** List of antibodies used in this study. Rb: rabbit; Ms: mouse; IP: immunoprecipitation; ChIP: chromatin immunoprecipitation; IB: immunoblot; PLA: proximity ligation assay.

Target	Ref. (Company)	Species	Experiments
Progesterone Receptor (PR)	sc-7208 (SCBT)	Rb	IP/ChIP/IB
PR	MA1-12626 (ThermoF)	Ms	PLA/IF
PRMT1	P1620 (Millipore)	Ms	IB/PLA
PRMT1	#07-404 (Millipore)	Rb	IB/PLA
PRMT1	#A300-722A (Bethyl)	Rb	IP/ChIP
HP1 gamma (HP1γ)	ab10480 (Abcam)	Rb	IP
HP1γ	ab66617 (Abcam)	Rb	IB/PLA
LSD1	sc-53875 (SCBT)	Ms	IB/PLA
H3K9me3	#39161 (ActiveMotif)	Rb	PLA
BRCA1	#OP92 (Millipore)	Ms	PLA
Tubulin	T6074 (Sigma)	Ms	IB

**Table 2 ijms-23-09509-t002:** List of primers used in this study. qPCR: quantitative polymerase chain reaction; ChIP: chromatin immunoprecipitation.

Target Gene	Experiment	Forward Sequence	Reverse Sequence
		(5′–3′)	(5′–3′)
*PRMT1*	qPCR	CGCCTCTTGAAGAAGTGTCCT	GATGCCAAAGTGTGCGTAGG
*EGFR*		GACAGGCCACCTCGTCG	CCGGCTCTCCCGATCAATAC
*STAT5A*		AAGCCCCACTGGAATGATGG	GGAGTCAAACTTCCAGGCGA
*BCL-X*		CCATCCACTCTACCCTCCCA	GTGTGGGGGTCTCACAGAA
*CCND1*		AAGCTCAAGTGGAACCT	AGGAAGTTGTTGGGGC
*GREB1*		CAAAGAATAACCTGTTGGCCCTGC	GACATGCCTGCGCTCTCATACTTA
*PDK4*		CATACTCCACTGCACCAACG	AGAAATTGGCAAGCCGTAAC
*BRCA1*		GTTGTTATGAAAACAGATGCTGAGTTTGTG	CTGGGTCACCCAGAAATAGCTAAC
*28S*		CGATCCATCATCCGCAATG	AGCCAAGCTCAGCGCAAC
*EGFR*	ChIP	[13]	
*BCL-X*			
*STAT5A*			
hChr1 (NG)		CGGGGGTCTTTTTGGACCTT	GAAACACGGCTGCCAGAAAC

## Data Availability

All unique reagents generated in this study (primers, plasmids) are available upon request from the Lead Contact. RNA sequencing data were submitted to Gene Expression Omnibus (GEO) and are available with the GSE134194 submission number. The Figure 4C was created with BioRender.com (with a publication license).

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
