# Peer review of "PRMT1, a Key Modulator of Unliganded Progesterone Receptor Signaling in Breast Cancer"

_ijms, 2022, doi:10.3390/ijms23179509_

Round 1
Reviewer 1 Report
This manuscript extends previous work of the authors investigating PRMT1 activity in breast cancer cells, in particular its role in unliganded progesterone receptor (PR) function. In general, the manuscript is well-structured and carefully prepared. English language is excellent, and the text is concise but with sufficient detail to understand the design and outcome of experiments as well as the authors’ interpretation of results. Figures are also clear and well-presented, and referencing is appropriate. Apart from several minor comments listed below, I feel that this manuscript is suitable for publication.
1. A major limitation of this study is that it has only been performed in a single cell line – T47D. Although frequently used for the study of PR, the relative expression of steroid hormone receptors (ER, PR, AR) in T47-D cells is quite unusual and results are sometimes not able to be replicated in other cell lines or are not reflected in human breast tumours. As this work progresses, I hope that at least some of the results are replicated in other PR-expressing human breast cancer cell lines, however in the present instance, it would be appropriate to add a few sentences in the Discussion section of the manuscript describing the major limitations of the study.
2. Have the authors performed any cell-free binding studies to indicate protein-protein interactions between PRMT1 and members of the HP1g-LSD1-uPR complex? This would seem to be a logical step to indicate whether PRMT1 association with this complex is solely due to its interaction with uPR (lines 106-109). Is it conceivable that PRMT1 affects the stability of the HP1g-LSD1-uPR complex?
3. Does MS 023 alter PRMT1 protein levels? (This is to confirm interpretation of results presented in Figure 2E).
4. It is noticeable that siPRMT1 only partially decreases PRMT1 levels. How might this affect interpretation of results?
5. Minor errors: (i) Line 207: ‘deletion’ should be ‘depletion’, (ii) Line 250: ‘ubiquitine’ should be ‘ubiquitin’, (iii) Line 258: ‘in fine’ (? no suggestion, unknown meaning). (iv) could “reminiscent of BRCA1 silencing” be changed to more scientific language, (v) Line 376: ‘ThermoFisher’, (vi) Line 388: ‘was incubated’, (vii) Line 416: ‘randomly’ should be ‘random’, (viii) Line 433: ‘5x106 starved cells’
Author Response
Reviewer 1
This manuscript extends previous work of the authors investigating PRMT1 activity in breast cancer cells, in particular its role in unliganded progesterone receptor (PR) function. In general, the manuscript is well-structured and carefully prepared. English language is excellent, and the text is concise but with sufficient detail to understand the design and outcome of experiments as well as the authors’ interpretation of results. Figures are also clear and well-presented, and referencing is appropriate. Apart from several minor comments listed below, I feel that this manuscript is suitable for publication.
- A major limitation of this study is that it has only been performed in a single cell line – T47D. Although frequently used for the study of PR, the relative expression of steroid hormone receptors (ER, PR, AR) in T47-D cells is quite unusual and results are sometimes not able to be replicated in other cell lines or are not reflected in human breast tumors. As this work progresses, I hope that at least some of the results are replicated in other PR-expressing human breast cancer cell lines, however in the present instance, it would be appropriate to add a few sentences in the Discussion section of the manuscript describing the major limitations of the study.
Response:
We totally understand the point raised here. We used a breast cancer (BC) cell line (i.e., T47D) with a phenotype particularly useful to dissect the regulatory steps of progesterone receptor (PR) activity. Most of the mechanistic works on PR have been done in T47D cells because they express much higher constitutive levels of PR, compared to other BC cells. This explains on the one hand, why all the available studies on the unconventional, unliganded PR pathway, have focused on T47D cells, including the pioneer work done by Beato and colleagues (using the modified T47D-MTVL cells), for obvious signal optimization reasons (Vicent, GP. et al., Genes Dev, 2013). On the other hand, conflicting and inconsistent reports regarding PR status or progestin response in other luminal A breast cancer cell lines, including MCF-7, make it difficult to work on PR signaling in another established BC cell line (Kao, J. et al., Plos One, 2009; Yu, S. et al., Biochem Biophys Res Commun. 2017- PMID: 28342866). However, being able to determine the clinical relevance of our in vitro mechanistic findings in a more physiological preclinical model, is one of our priorities, notably by using mouse and patient-derived xenografts. Some of these elements were added in the discussion section (from line 375).
- Have the authors performed any cell-free binding studies to indicate protein-protein interactions between PRMT1 and members of the HP1g-LSD1-uPR complex? This would seem to be a logical step to indicate whether PRMT1 association with this complex is solely due to its interaction with uPR (lines 106-109).
Response
We want to thank the reviewer for this interesting suggestion. We did not perform cell-free binding assays to study interactions between PRMT1 and LSD1 or HP1g. However, we confirmed these interactions with two different assays in cellulo (Coimmunoprecipitation and Proximity Ligation Assays). We therefore cannot conclude if these interactions are direct or not. We modified figure 4C according to this comment, by removing the “direct” interactions between PRMT1 and these actors of the complex.
Some element of response can be provided by a recent report that demonstrated a direct interaction between LSD1 and CARM1/PRMT4 (Liu, J et al., EMBO Rep. 2020), which share strong structural homologies with PRMT1 (Hwang, JW. et al., Exp Mol Med, 2021; Malbeteau, L. et al., Endocr Rev. 2022).
-Is it conceivable that PRMT1 affects the stability of the HP1g-LSD1-uPR complex?
Response:
We think it is conceivable to think that PRMT1 could affect the stability of the repressive complex, since we observed more PR (unliganded) upon PRMT1 loss in the absence of progesterone (Fig. 1H, 2C, 4B). This increase in PR expression could reflect either more PR proteins or more stabilized receptors, both cases could definitely affect the structure and/or the stability of the HP1g-LSD1-uPR complex.
Moreover, our preliminary PLA data analyzing PR/HP1g interactions upon siPRMT1 in T47D cells tend to show that targeting PRMT1 by siRNA increases the interaction.
- Does MS 023 alter PRMT1 protein levels? (This is to confirm interpretation of results presented in Figure 2E).
Response
To answer to the reviewer’s comment, we added this result in Figure 2E showing that the MS 023 inhibitor (60 nM) does not affect PRMT1 protein levels.
- It is noticeable that siPRMT1 only partially decreases PRMT1 levels. How might this affect interpretation of results?
Response
We agree with this observation, the knockdown of PRMT1 is only partial. Nevertheless, we observed significant effects upon siPRMT1 that are rescued when we added PRMT1-Flag (Rescue experiment, fig. 3D), confirming the specificity of PRMT1 action and ruling out the possibility of off-target effects. Moreover, the different PLAs assessing interactions between PRMT1 and partners in cells transfected with siPRMT1 all showed significant decreases in the number of dots (=interactions) (Fig. 1), supporting a strong PRMT1 KD. Nevertheless, as PRMT1 is essential in cells and mice, notably by maintaining genome integrity, it is preferable to perform a moderate PRMT1 decrease.
- Minor errors: (i) Line 207: ‘deletion’ should be ‘depletion’, (ii) Line 250: ‘ubiquitine’ should be ‘ubiquitin’, (iii) Line 258: ‘in fine’ (? no suggestion, unknown meaning). (iv) could “reminiscent of BRCA1 silencing” be changed to more scientific language, (v) Line 376: ‘ThermoFisher’, (vi) Line 388: ‘was incubated’, (vii) Line 416: ‘randomly’ should be ‘random’, (viii) Line 433: ‘5x106 starved cells’
Response:
We thank the reviewer for highlighting these typos and have now changed the text accordingly.

Reviewer 2 Report
I thank the academic editor for allowing me to review this very interesting manuscript in which the authors conduct, using breast carcinoma cell lines and using siRNAs, an elegant investigation aimed at demonstrating how important the PRMT1 protein is in intracellular signaling dependent on the receptor. unligated progesterone. I believe this manuscript deserves publication in IJMS. I suggest the authors correct some typoos. Congratulations.
Author Response
Reviewer 2
I thank the academic editor for allowing me to review this very interesting manuscript in which the authors conduct, using breast carcinoma cell lines and using siRNAs, an elegant investigation aimed at demonstrating how important the PRMT1 protein is in intracellular signaling dependent on the receptor. unligated progesterone. I believe this manuscript deserves publication in IJMS. I suggest the authors correct some typos. Congratulations.
Response
The text went through a final round of English proofreading by a native English speaker.

Reviewer 3 Report
Topic of manusript is interesting and give important finding for the breast cancer biology. The methodological part of the manuscript is described sufficiently and it is sufficient to substantiate the presented results. I have only two question for authors.
-
Could shourtly introduce and discuss usebility and potential benefit of presented model in therapy design?
-
In vitro study was done by using ERalfa positive cells. But role ERalfa is described only shortly. Could be possible discuss more? Could be also mentioned, or shortly discuss role of ERbeta in presented model?
Author Response
Reviewer 3
Topic of manusript is interesting and give important finding for the breast cancer biology. The methodological part of the manuscript is described sufficiently and it is sufficient to substantiate the presented results. I have only two question for authors.
- Could shortly introduce and discuss useability and potential benefit of presented model in therapy design?
Response
It is now admitted that PR influences cell tumorigenesis in the absence of ligands. For instance, in unstimulated breast cells, unliganded PRs (uPRs) have been shown to be primarily growth suppressive (Daniel, AR et al., Oncogene, 2015; Zheng ZY, et al., Breast Cancer Res Treat, 2008). Our work and others propose to link this phenotype to the repression of genes dependent on uPR (Vicent, GP. et al., Genes Dev, 2013). We believe that keeping those specific genes in a silent state prior to hormonal activation is essential for breast cells, to limit the development of cancer. These new findings raise exciting questions about the changes that could operate after menopause, when ovarian function ceases and hormonal levels fall substantially to near zero. We can speculate that the greater risk of breast cancer observed in postmenopausal women with hormonal replacement therapy (HRT) containing progestogen, compared to preparations containing estrogen alone (Collaborative Group on Hormonal Factors in Breast Cancer., Lancet, 2019), could be, at least in part, dependent on the unexpected and still underestimated effects of the unliganded PR signaling. The hormonal drop in postmenopausal BC patients might reinforce the uPR-dependent gene silencing, and thus protective effects against cell proliferation, that are reversed when progestogens are added through HRT. Since the impact of HRT may vary from one woman to another after menopause, investigating the status/activation of unliganded PR signaling may offer promising new diagnostic avenues for preventing the development of BC.
Moreover, our model suggests that PRMT1 is an essential player in controlling PR protein level and expression of target genes. Interestingly, PRMT1 has been shown to be dramatically overexpressed in breast cancer, and patients with higher PRMT1 expression show a higher malignancy grade, partly through the activation of proliferation-related genes (Suresh, S et al., Cancers (Basel), 2022; Liu, LM et al., Cancer Res, 2019). Our findings raise the idea that a higher expression of PRMT1 in breast cancer patients, especially those experiencing hormonal drop, could be associated with PR instability and downstream aberrant expression of proliferation-related genes, leading to the neutralization/inversion of the protective and anti-tumoral effect offered by uPR-dependent gene repression/reprogramming.
Some of these points were added to the discussion (from line 368 onwards).
- In vitro study was done by using ERa positive cells. But role ERa is described only shortly. Could be possible discuss more? Could be also mentioned, or shortly discuss role of ERbeta in presented model?
Response
In this study, our objective was to investigate PR signaling (as mentioned in the discussion), however, we are aware that important cross-talks between ERa and PR pathways exist, both at the genomic and protein levels, that could deeply influence BC outcome. So far, except for the fact that ERβ expression is an independent prognostic marker in ERα+/PR+ breast cancers, we did not find any specific information on the role of ERb in PR signaling, liganded or not, and divergent literature exists about the role of this isoform b in breast cancer cell lines, especially in T47D cells (Mal, R et al., Front Oncol, 2020).
However, we seized the opportunity here to highlight a recent paper demonstrating that unliganded PR (uPR) can govern ESR1 expression by regulating DNA methylation (Verde, G et al., Cancers (Basel), 2018). The binding of uPR to the low-methylated ESR1 promoter maintains ESR1 basal expression, and thus ERa-dependent activity. However, depletion of PR is associated with increased DNA methylation at the ESR1 promoter and therefore less ERa expression. This phenotype is irreversible, the high levels of methylation at the ESR1 promoter in PR-depleted cells persist after its re-expression and impede PR binding to the promoter. This work clearly highlights that unliganded PR is a key determinant in ERa expression and could definitely influence how cancer cells respond to estrogen and selective ER modulators (SERM). As PGR is a target gene of ERa, endocrine therapies targeting ERa necessarily result in the loss of PR expression. Indeed, investigating the role of PR in hormone-free breast cancer cells is crucial to understand how cancer cells may respond to treatment. We know for instance that ER+/PR− breast cancers are less likely to respond to SERM therapy than ER+/PR+ BCs, thus understanding how unliganded PR may affect this will help to design more appropriate targeted therapies for breast cancer patients.
Some of these elements were added in the discussion section (from line 339 onwards).

Reviewer 4 Report
Dear Authors:
The manuscript by Malbeteau et al has demonstrated the role of PRMT1 in breast cancer I have just a few suggestions.
1) I suggest to put material and method section before discussion section, and add conclusion section at the end.
2) Some background information or references are missing: Please add more background information about breast cancer (please cite: 1.https://www.frontiersin.org/articles/10.3389/fonc.2022.820968/abstract Front. Oncol., 22 June 2022 Sec.Breast Cancer https://doi.org/10.3389/fonc.2022.820968
2.Mitochondrial mutations and mitoepigenetics: Focus on regulation of oxidative stress-induced responses in breast cancers. Semin Cancer Biol. 2022 Aug;83:556-569. doi: 10.1016/j.semcancer.2020.09.012. Epub 2020 Oct 6. PMID: 33035656.)
Best,
Author Response
Reviewer 4
The manuscript by Malbeteau et al has demonstrated the role of PRMT1 in breast cancer. I have just a few suggestions.
- I suggest to put material and method section before discussion section and add conclusion section at the end.
Response:
We followed the guidelines from IJMS requiring that the material and methods section appears at the end of the paper, after the discussion section.
- 2. Some background information or references are missing. Please add more background information about breast cancer. Please cite:
- Advances in the Prevention and Treatment of Obesity-Driven Effects in Breast Cancers / PMID: 35814391
- Mitochondrial mutations and mitoepigenetics: Focus on regulation of oxidative stress-induced responses in breast cancers / PMID: 33035656
Response:
We would like to thank the reviewer for pointing out these interesting papers. However, they do not fall within the topic of our manuscript, and we decided not to add them to the manuscript.

Reviewer 5 Report
The authors present a manuscript entitled PRMT1, a key modulator of unliganded progesterone receptor 2 signaling in breast cancer.
The authors suggest PRMT1 is a novel actor in uPR signaling because in unstimulated T47D breast cancer cells, PRMT1 interacts and functions alongside uPR and its partners to target endogenous progesterone-responsive promoters
The manuscript indicates several advances in PRMT1 biology, yet this reviewer has several points of concern that requires attention prior to acceptance of said manuscript.
In experiment 1, the authors cultured cell in steroid-free media, and then in serum-free media. Have the authors made any remarks regarding serum free or steroid free alone conditions wit repot to PRMT1 Expression to interact with PR
-all siRNA experiments requires two or more siRNAs.
Figure 3 indicates downstream target of siPRMT1. The reviewer is concerned regarding the Venn diagram of R5020 regulated genes that is determined by nonspecific control siRNA. In this experiment, an additional control siRNA should be used, and approximately 156 shared gene target should still be present.
In figure 3, please show a more effective diagram of go-term targets, and or provide as supplemental data the remainder of said Goterm list. Also I am not seeing the GSEA analysis in figure 3.
In figure 4, can the authors indicate whether the abundance of BRCA1 is altered when siPR is transfected into T47Ds? That could alter the interpretation of the PLA assay in 4A,B,
Finally the diagram in figure 4c is speculation. The authors show no evidence of PRMT regulating LSD1, HP1y, etc.
Have the authors performed any experiment that indicates altered BRCA1 function upon serum/hormone depletion and/or siRNA treatment?
Author Response
Reviewer 5
The authors present a manuscript entitled PRMT1, a key modulator of unliganded progesterone receptor 2 signaling in breast cancer. The authors suggest PRMT1 is a novel actor in uPR signaling because in unstimulated T47D breast cancer cells, PRMT1 interacts and functions alongside uPR and its partners to target endogenous progesterone-responsive promoters. The manuscript indicates several advances in PRMT1 biology, yet this reviewer has several points of concern that requires attention prior to acceptance of said manuscript.
- In experiment 1, the authors cultured cell in steroid-free media, and then in serum-free media. Have the authors made any remarks regarding serum free or steroid free alone conditions wit repot to PRMT1 Expression to interact with PR?
Response:
We aimed to reproduce the culture conditions used in the pioneer publication on the role of unliganded PR in breast cancer cells (Vicent, GP. et al., Genes Dev, 2013); these stringent starvation conditions (RPMI medium with 10% dextran-coated charcoal-treated FBS for 48 h and in serum-free medium for 24 h) allowed us to work in favorable conditions to study uPR signaling, excluding possibilities to have ligand (other than steroids) able to bind to the receptors, but also to better mimic a hormonal-deficient condition like a postmenopausal stage.
We performed several tests analyzing the expression of PRMT1 under the two different culture conditions (steroid-free or total serum-free media) and observed no significant differences (see WB).
- All siRNA experiments requires two or more siRNAs.
Response:
As mentioned in the Material and Methods section, we used a pool of 3 or 4 siRNAs (according to the manufacturer) to target the proteins of interest.
- Figure 3 indicates downstream target of siPRMT1. The reviewer is concerned regarding the Venn diagram of R5020 regulated genes that is determined by nonspecific control siRNA. In this experiment, an additional control siRNA should be used, and approximately 156 shared gene targets should still be present.
Response:
As also previously described in Malbeteau et al., iScience (2020), we largely validated the specificity of our results by doing qPCRs on random genes regulated in the whole-genome sequencing, to confirm that our non-specific siRNAs (siNS) did not affect gene expression in our study (see two examples attached).
- In figure 3, please show a more effective diagram of go-term targets, and or provide as supplemental data the remainder of said Goterm list. Also, I am not seeing the GSEA analysis in figure 3.
Response
We want to thank the reviewer for his/her pertinent comment. Indeed, we did not perform a GSEA but rather a gene ontology analysis (using GSEA software) to determine whether the genes basally regulated by PRMT1 in unstimulated cells belong to specific signaling pathways that might be relevant to the literature. We accordingly changed the term GSEA in the paper (legend of figure 3E).
We also modified the figure 3E by inserting the whole tab from this analysis, as the reviewer suggested.
- In figure 4, can the authors indicate whether the abundance of BRCA1 is altered when siPR is transfected into T47Ds? That could alter the interpretation of the PLA assay in 4 A,B.
Response:
When siPR is transfected into cells, the PR-BRCA1 interaction decreases, which makes sense because PR is targeted (Figure 4A). This result would be the same even if BRCA1 expression was affected. We did not analyze BRCA1 expression after PR KD, because it does not seem relevant here, the PR-BRCA1 association will still be decreased as long as PR is knocked down. However, conversely, Beato and colleagues previously showed that BRCA1 down-regulation/KD induces increased expression of liganded and unliganded PR in T47D cells (Calvo, V. and Beato, M., Cancer Res, 2011). This result is exciting because it suggests that BRCA1 could be the E3-ubiquitin ligase responsible for unliganded PR turnover and thus, a key factor in the regulation of PR-target genes in the absence of ligands.
- Finally, the diagram in figure 4c is speculation. The authors show no evidence of PRMT1 regulating LSD1, HP1y, etc.
Response:
We agree with the reviewer that we did not have direct evidence showing that PRMT1 regulates LSD1 or HP1g. We demonstrated that PRMT1 is part of the HP1g-LSD1-uPR complex and participates in the repression of uPR-target genes. These two main conclusions are illustrated in our Figure 4C. However, we do not know yet if this is dependent or not on LSD1 and/or HP1g regulation (direct or indirect). We have thus modified the figure 4C accordingly, by removing the “direct” interactions between PRMT1 and these actors of the complex.
- Have the authors performed any experiment that indicates altered BRCA1 function upon serum/hormone depletion and/or siRNA treatment?
We thank the reviewer for his/her suggestion. We did not perform any experiments to analyze alterations of BRCA1 activity upon treatment. From our point of view, this could be the topic of a future article, since interesting findings have highlighted some correlation between BRCA1 and PR signaling in breast cancer (Lee, O. et al., Cancer Lett, 2021; Ma, Y. et al., Mol Endocrinol, 2006).

Round 2
Reviewer 4 Report
Suggest for publication
Author Response
We thank the reviewer for his/her postive comments.
English has been corrected as requested.
Reviewer 5 Report
The authors present a manuscript entitled PRMT1, a key modulator of unliganded progesterone receptor 2 signaling in breast cancer. The authors suggest PRMT1 is a novel actor in uPR signaling because in unstimulated T47D breast cancer cells, PRMT1 interacts and functions alongside uPR and its partners to target endogenous progesterone-responsive promoters. The manuscript indicates several advances in PRMT1 biology, yet this reviewer has several points of concern that requires attention prior to acceptance of said manuscript.
- In experiment 1, the authors cultured cell in steroid-free media, and then in serum-free media. Have the authors made any remarks regarding serum free or steroid free alone conditions wit repot to PRMT1 Expression to interact with PR?
Response:
We aimed to reproduce the culture conditions used in the pioneer publication on the role of unliganded PR in breast cancer cells (Vicent, GP. et al., Genes Dev, 2013); these stringent starvation conditions (RPMI medium with 10% dextran-coated charcoal-treated FBS for 48 h and in serum-free medium for 24 h) allowed us to work in favorable conditions to study uPR signaling, excluding possibilities to have ligand (other than steroids) able to bind to the receptors, but also to better mimic a hormonal-deficient condition like a postmenopausal stage.
We performed several tests analyzing the expression of PRMT1 under the two different culture conditions (steroid-free or total serum-free media) and observed no significant differences (see WB).
RESPONSE: OKAY
- All siRNA experiments requires two or more siRNAs.
Response:
As mentioned in the Material and Methods section, we used a pool of 3 or 4 siRNAs (according to the manufacturer) to target the proteins of interest.
RESPONSE: OKAY
- Figure 3 indicates downstream target of siPRMT1. The reviewer is concerned regarding the Venn diagram of R5020 regulated genes that is determined by nonspecific control siRNA. In this experiment, an additional control siRNA should be used, and approximately 156 shared gene targets should still be present.
Response:
As also previously described in Malbeteau et al., iScience (2020), we largely validated the specificity of our results by doing qPCRs on random genes regulated in the whole-genome sequencing, to confirm that our non-specific siRNAs (siNS) did not affect gene expression in our study (see two examples attached).
RESPONSE: Please provide such validation data in suppl. data.
- In figure 3, please show a more effective diagram of go-term targets, and or provide as supplemental data the remainder of said Goterm list. Also, I am not seeing the GSEA analysis in figure 3.
Response
We want to thank the reviewer for his/her pertinent comment. Indeed, we did not perform a GSEA but rather a gene ontology analysis (using GSEA software) to determine whether the genes basally regulated by PRMT1 in unstimulated cells belong to specific signaling pathways that might be relevant to the literature. We accordingly changed the term GSEA in the paper (legend of figure 3E).
We also modified the figure 3E by inserting the whole tab from this analysis, as the reviewer suggested.
RESPONSE: OKAY
- In figure 4, can the authors indicate whether the abundance of BRCA1 is altered when siPR is transfected into T47Ds? That could alter the interpretation of the PLA assay in 4 A,B.
Response:
When siPR is transfected into cells, the PR-BRCA1 interaction decreases, which makes sense because PR is targeted (Figure 4A). This result would be the same even if BRCA1 expression was affected. We did not analyze BRCA1 expression after PR KD, because it does not seem relevant here, the PR-BRCA1 association will still be decreased as long as PR is knocked down. However, conversely, Beato and colleagues previously showed that BRCA1 down-regulation/KD induces increased expression of liganded and unliganded PR in T47D cells (Calvo, V. and Beato, M., Cancer Res, 2011). This result is exciting because it suggests that BRCA1 could be the E3-ubiquitin ligase responsible for unliganded PR turnover and thus, a key factor in the regulation of PR-target genes in the absence of ligands.
RESPONSE: Determining the level of BRCA1 after siPR would actually be of interest with regards to understanding the regulator of BRCA1 in postmenopausal scenarios.
- Finally, the diagram in figure 4c is speculation. The authors show no evidence of PRMT1 regulating LSD1, HP1y, etc.
Response:
We agree with the reviewer that we did not have direct evidence showing that PRMT1 regulates LSD1 or HP1g. We demonstrated that PRMT1 is part of the HP1g-LSD1-uPR complex and participates in the repression of uPR-target genes. These two main conclusions are illustrated in our Figure 4C. However, we do not know yet if this is dependent or not on LSD1 and/or HP1g regulation (direct or indirect). We have thus modified the figure 4C accordingly, by removing the “direct” interactions between PRMT1 and these actors of the complex.
RESPONSE: OKAY
- Have the authors performed any experiment that indicates altered BRCA1 function upon serum/hormone depletion and/or siRNA treatment?
We thank the reviewer for his/her suggestion. We did not perform any experiments to analyze alterations of BRCA1 activity upon treatment. From our point of view, this could be the topic of a future article, since interesting findings have highlighted some correlation between BRCA1 and PR signaling in breast cancer (Lee, O. et al., Cancer Lett, 2021; Ma, Y. et al., Mol Endocrinol, 2006).
RESPONSE: OKAY
Author Response
Reviewer 5
- Figure 3 indicates downstream target of siPRMT1. The reviewer is concerned regarding the Venn diagram of R5020 regulated genes that is determined by nonspecific control siRNA. In this experiment, an additional control siRNA should be used, and approximately 156 shared gene targets should still be present.
Response:
As also previously described in Malbeteau et al., iScience (2020), we largely validated the specificity of our results by doing qPCRs on random genes regulated in the whole-genome sequencing, to confirm that our non-specific siRNAs (siNS) did not affect gene expression in our study (see two examples attached).
RESPONSE: Please provide such validation data in suppl. data.
Response to reviewer.
We added a sentence to refer to our previous article where the controls have already been done. As we don’t have any supplemental material, we don’t think it is worth creating one for this result and it is not justified to put it in the main manuscript.
- In figure 4, can the authors indicate whether the abundance of BRCA1 is altered when siPR is transfected into T47Ds? That could alter the interpretation of the PLA assay in 4 A,B.
Response:
When siPR is transfected into cells, the PR-BRCA1 interaction decreases, which makes sense because PR is targeted (Figure 4A). This result would be the same even if BRCA1 expression was affected. We did not analyze BRCA1 expression after PR KD, because it does not seem relevant here, the PR-BRCA1 association will still be decreased as long as PR is knocked down. However, conversely, Beato and colleagues previously showed that BRCA1 down-regulation/KD induces increased expression of liganded and unliganded PR in T47D cells (Calvo, V. and Beato, M., Cancer Res, 2011). This result is exciting because it suggests that BRCA1 could be the E3-ubiquitin ligase responsible for unliganded PR turnover and thus, a key factor in the regulation of PR-target genes in the absence of ligands.
RESPONSE: Determining the level of BRCA1 after siPR would actually be of interest with regards to understanding the regulator of BRCA1 in postmenopausal scenarios.
Response to the reviewer :
As suggested, we did analyze the expression of BRCA1 upon siPR (by qPCR) and our result shows a decrease in BRCA1 gene expression, in starved T47D cells (graph attached). We did not have any complementary approach to confirm this tendency but based on these two independent experiments, we could extrapolate that the use of selective estrogen receptor modulators/degraders (SERMs/SERDs) that were shown to induce a marked reduction in PGR levels in women, could impact the expression of the BRCA1 tumor suppressor gene, and be harmful for patients. However, post-menopausal transition leads to a drop of hormones but, less circulating hormones does not necessarily result in less receptor expression, so definitely more work is needed to be able to conclude anything on the regulation of BRCA1 after menopause in breast cancer patients.
